# Excessive Reasoning Attack on Reasoning LLMs

## Abstract

Recent reasoning large language models (LLMs), such as OpenAI o1 and DeepSeek-R1, exhibit strong performance on complex tasks through test-time inference scaling. However, prior studies have shown that these models often incur significant computational costs due to excessive reasoning, such as frequent switching between reasoning trajectories (e.g., underthinking) or redundant reasoning on simple questions (e.g., overthinking). In this work, we expose a novel threat: crafting adversarial inputs to exploit excessive reasoning behaviors. However, directly optimizing for excessive reasoning is non-trivial because reasoning length is non-differentiable. To overcome this, we introduce a proxy framework that approximates the long reasoning objective and shapes token-level behavior: (1) Priority Cross-Entropy Loss, a modification of the standard cross-entropy objective that emphasizes key tokens by leveraging the autoregressive nature of LMs; (2) Excessive Reasoning Loss, which encourages the model to initiate additional reasoning paths during inference; and (3) Delayed Termination Loss, which is designed to extend the reasoning process and defer the generation of final outputs. We optimize and evaluate our attack for the GSM8K and ORCA datasets on DeepSeek-R1-Distill-LLaMA and DeepSeek-R1-Distill-Qwen. Empirical results demonstrate a 3x to 6.5x increase in reasoning length with comparable utility performance. Furthermore, our crafted adversarial inputs exhibit transferability, inducing computational overhead in o3-mini, GPT-OSS, DeepSeek-R1, and QWQ models. Our findings highlight an emerging efficiency-oriented vulnerability in modern reasoning LLMs, posing new challenges for their reliable deployment.

## 1 Introduction

Inference-time scaling has emerged as a critical technique for enhancing the reasoning capabilities of LLMs (Wei et al., 2022; Wang et al., 2023; Besta et al., 2024; Yang et al., 2024). Methods such as Chain-of-Thought (CoT) (Wei et al., 2022) guide models through intermediate reasoning steps by outside prompts, while recent models like DeepSeek (DeepSeek-AI et al., 2025) and QWQ (Team, 2025) embed reasoning capabilities inside models themselves during training. However, recent studies have shown that these models often suffer from excessive reasoning behaviors, such as frequent shifts in reasoning strategies or redundant processing, which can lead to substantial computational overhead (Wang et al., 2025; Chen et al., 2024). In addition, recent studies (Turpin et al., 2023; Wu et al., 2025) have shown that longer reasoning chains can sometimes result in unfaithful or incorrect outputs. In particular, Wu et al. (2025) demonstrates that task accuracy initially improves but declines once the reasoning becomes too lengthy. These inefficiencies introduce a novel security risk: the attacker can craft adversarial inputs to exploit them, intentionally inflating inference-time resource usage or inducing unfaithful behaviors.

Prior work has explored related threats in LMs. For example, Sponge examples (Shumailov et al., 2021) increase computational costs by maximizing activation norms, while the NICGSlowdown attack (Chen et al., 2022) manipulates token logits to delay output generation. More recently, Kumar et al. (2025) propose the Overthink attack, which adopts an indirect prompt injection strategy and inserts a decoy to external resources. This compels the model to allocate additional reasoning resources toward solving an intermediary task before addressing the primary query. These attacks highlight the vulnerability of LMs to resource-exhaustion threats. However, these approaches either

rely on modifying model internals, controlling output logits, or injecting external distractions, which can limit their generality and practicality.

In this work, we aim to directly perturb the input to elicit excessive reasoning behavior, increasing computational overhead without requiring external content or architectural manipulation. However, directly optimizing for excessive reasoning is non-trivial because reasoning length is non-differentiable explicitly. Therefore, we propose differentiable proxy losses to approximate the length objective and enhance the attack performance via shaping token-level behavior. Concretely, we craft the adversarial suffixes with three complementary losses:

- **Priority Cross-Entropy Loss** prioritizes key tokens while masking less informative ones to enhance optimization efficiency. This loss leverages the autoregressive nature of LM to enable more targeted and effective gradient updates.

- **Excessive Reasoning Loss** increases the likelihood of branched or recursive reasoning, leading to greater computational overhead.

- **Delayed Termination Loss** encourages the model to defer the termination of reasoning and answer generation.

We optimize and evaluate our attacks for the GSM8K (Cobbe et al., 2021) and ORCA (Mitra et al., 2024) datasets on DeepSeek-R1-Distill-Llama and DeepSeek-R1-Distill-Qwen. Our attacks consistently increase the reasoning length by over 3x to 6.5x using only 10 crafted adversarial tokens. Moreover, our attack demonstrates strong transferability across models on commercial platforms, including OpenAI o3-mini, GPT-OSS (Jaech et al., 2024), DeepSeek-R1, and QWQ, suggesting a broader vulnerability among reasoning-optimized LLMs. These findings expose an underexplored issue. While such models are proficient in reasoning, they remain susceptible to targeted manipulations that exploit their reasoning mechanisms to induce significant computational overhead. Our results underscore the urgent need for defenses that can detect and mitigate excessive reasoning triggered by adversarial prompts.

## 2 METHODOLOGY

This section presents our adversarial attack framework that increases the computational overhead of reasoning LLMs by inducing excessive reasoning. We begin by stating the threat model and the use cases. We then conduct two studies. First, we design proxy objectives to approximate the non-differentiable length objective. Second, we introduce two token-level losses that further shape token behaviors. Finally, we describe the optimization procedure used to generate adversarial attacks.

### 2.1 THREAT MODEL

The primary objective of our attack is to craft inputs (e.g., suffixes) that compel the model to extend the reasoning processes as long as possible, thus significantly increasing the computational cost at inference time. Similar to prior work Carlini et al. (2023); Zou et al. (2023); Gao et al. (2024); Boucher et al. (2022), we assume a white-box scenario in which the attacker has complete access to the model's architecture, parameters, and gradients.

Following Carlini et al. (2023), we consider two primary use cases for our attack. In the first use case, a malicious user intentionally induces excessive computational load, degrading overall system performance and diminishing service quality for other users, akin to a DoS attack. In the second use case, a benign user queries the model within an autonomous system (e.g., LLM Agent) that processes untrusted third-party data (e.g., crafted adversarial data), resulting in significantly higher costs (e.g., money) than expected. As we later demonstrate, these crafted adversarial inputs exhibit strong transferability across different models. Moreover, our attack aligns with the Model Denial of Service (MDoS) threat as defined by OWASP, wherein adversarial inputs lead to resource exhaustion, degrading system responsiveness and service availability for other users.[1]

---

[1] https://genai.owasp.org/llmrisk2023-24/llm04-model-denial-of-service/

## 2.2 APPROXIMATE OBJECTIVE FOR LONG REASONING TRAJECTORIES

In this attack, we aim to craft inputs that elicit the model to produce excessive reasoning. Formally, this corresponds to optimizing the objective $\mathcal{L}_{long}$ that rewards generations exhibiting excessive reasoning. Since $\mathcal{L}_{long}$ depends on non-differentiable properties of sampled text (e.g., the length and density of intermediate steps), it cannot be optimized directly.

Following the adversarial prompt optimization paradigm, we instead optimize against a differentiable proxy. Prior work, such as ATA (Gan et al., 2024) and GCG (Zou et al., 2023), employs cross-entropy (CE) against short fixed targets (e.g., "Sorry, I'm unable to answer the question" or "Sure, I can...") as surrogates for latent objectives like refusal or compliance. A natural extension is to use CE with long, reasoning-dense targets as a proxy for $\mathcal{L}_{long}$. However, this direct approach is ineffective since when supervision is spread uniformly across thousands of tokens, gradients become diffuse. Moreover, because the adversarial suffix typically contains only a few tokens, forcing it to match the distribution of thousands of target tokens is unrealistic.

To address this, we propose Priority Cross-Entropy (PCE) loss, which reweights supervision to focus on informative, prompt-dependent tokens rather than treating all positions equally. We further enhance the attack performance by constructing reasoning-rich target sequences with DSPy (Khattab et al., 2024), yielding more effective surrogates for the $\mathcal{L}_{long}$ objective.

**Optimizing toward long targets.** We begin by examining the standard CE objective and its limitations when applied to a long target. Traditional adversarial attacks on LMs minimize CE to increase the likelihood of a target sequence (e.g., "Sure, I can ...."). Formally, for base input $x$, adversarial suffix $x'$, and target sequence $y$, the objective is

$$\mathcal{L}_{\text{CE}} = -\log p(y \mid \{x, x'\}). \tag{1}$$

In prior work, the target $y$ is typically short (often $< 10$ tokens). However, to elicit excessive reasoning, $y$ is much longer (e.g., $> 1{,}000$ tokens). Uniformly supervising all positions in such sequences is optimization-inefficient since the gradient signal is spread across thousands of tokens. Also, many tokens (e.g., "the" and "I") can be accurately generated even without the prompt, due to statistical priors learned during pretraining. This suggests that optimizing every token is unnecessary; instead, supervision should focus on the prompt-dependent tokens that may influence the emergence of excessive reasoning.

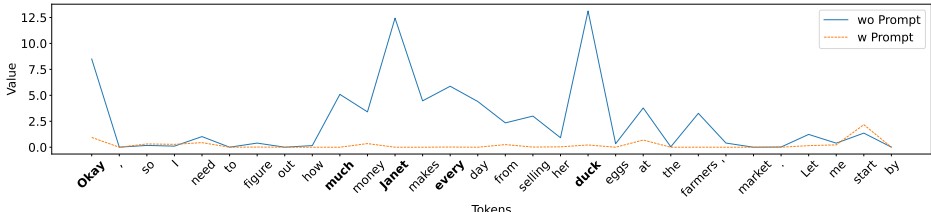

Figure 1: The perplexity of a reasoning sample in the output with and without the prompt. Bold tokens are assigned 1 in the mask.

To investigate this, we analyze the per-token perplexity distribution of a target sequence with and without the input prompt. As shown in Fig. 1, only a small subset of tokens exhibits a substantial change in perplexity when the prompt is removed. This supports our hypothesis that informativeness is not uniformly distributed across tokens, and that only a subset is highly dependent on the prompt.

Building on this insight, we design a token-level importance mask that emphasizes tokens the model considers informative, thereby improving optimization efficiency. For each target token $y_t$, we compute an importance score as the change in log-probability with and without the prompt:

$$S_t = \log p(y_t \mid x, y_{<t}) - \log p(y_t \mid y_{<t}). \tag{2}$$

This score quantifies the extent to which each token's prediction relies on the presence of the prompt. We then construct a binary mask $\mathcal{M}$ by selecting the top $K\%$ of tokens with the highest importance scores (set $M_t = 1$) and assigning zero weight to the rest (set $M_t = 0$).

**Priority Cross-Entropy Loss.** Putting these pieces together, we modify standard CE to focus supervision on prompt-sensitive tokens. The resulting PCE objective is

$$\mathcal{L}_{\text{PCE}} = -\frac{1}{|y|} \sum_{t=1}^{|y|} \mathcal{M}_t \cdot \log p(y_t \mid \{x, x'\}, y_{<t}). \tag{3}$$

This targeted supervision improves optimization efficiency and increases the likelihood that the model produces excessive reasoning at inference. Empirically, PCE substantially raises the frequency of reasoning (Fig. 3).

**Constructing long targets.** Because our attack seeks to elicit excessive reasoning, the proxy objective itself must involve long outputs. Short fixed targets (as in ATA and GCG) are less effective here since they do not encourage extended reasoning and provide little supervision on reasoning structure. We therefore construct long proxy targets. As simple baselines, we (i) sample multiple model outputs per input and select the longest sequence, and (ii) append the reasoning-oriented prompt (e.g., CoT) that naturally elicits step-by-step derivations to each input.

To obtain stronger proxies, we first leverage DSPy to iteratively refine the CoT prompt on a small dataset with the explicit objective of maximizing output length. Then, we query the model with each input and append the DSPy-optimized prompt to obtain the long, reasoning-dense target trajectories. Together with PCE, these long trajectories serve as effective proxies for the $\mathcal{L}_{long}$ objective. App. C.3 provides the DSPy-optimized prompt, a sample output, and statistics across different prompting baselines. We also examine the impact of various construction baselines in App. C.5.

## 2.3 Shaping Token-Level Behavior for Excessive Reasoning

Building on PCE and long targets, we next shape token-level behavior to further promote excessive reasoning. We introduce two complementary losses: Excessive Reasoning (ER) Loss, which increases the likelihood of reasoning-associated tokens, and Delayed Termination (DT) Loss, which discourages premature end-of-thinking/sequence decisions.

**Excessive Reasoning Loss.** In the previous section, we have described how DSPy-optimized prompts yield long target trajectories. Within these outputs, we observe that certain tokens (e.g., *"Wait"*, *"Alternatively"*; see App. C.3) frequently occur in outputs and often signal branching or recursive reasoning steps, consistent with prior findings (Wang et al., 2025; Chen et al., 2024). To exploit this behavior, we aim to increase the likelihood of generating such tokens during the reasoning. While it is possible to construct a manual list of indicative tokens, this approach does not scale and may miss less obvious but influential cases. Instead, we adopt a data-driven approach to automatically identify reasoning-associated tokens. As demonstrated in our analysis (Sec. 3.2), this approach uncovers influential tokens that would likely be overlooked through manual inspection, underscoring the efficacy of our method.

Concretely, we extract the top $n$ most frequent tokens that appear in the first two positions of each sentence generated during constructing long-from targets. These tokens are hypothesized to play a critical role in initiating new reasoning trajectories. Let $\mathcal{T}$ denote the collected set of high-impact tokens. To promote their occurrence during generation, we define the ER Loss as:

$$\mathcal{L}_{\text{ER}} = -\frac{1}{|y|} \sum_{t=1}^{|y|} \sum_{v \in \mathcal{T}} \log p(y_t = v \mid \{x, x'\}, y_{<t}). \tag{4}$$

This objective increases the likelihood of generating collected high-impact tokens, thereby inducing longer reasoning sequences.

**Delayed Termination Loss.** In many reasoning LLMs, the generation process typically begins with intermediate reasoning steps, which conclude with a designated end-of-thinking (EOT) token (e.g., `</think>`). Then, the model would generate an answer conclusion terminated by an end-of-sequence (EOS) token (e.g., `<eos>`). To prolong both the reasoning and answer conclusion phases, we aim to reduce the model's tendency to emit these termination tokens during decoding. However, due to the stochastic nature of autoregressive generation, the precise timestep at which these tokens appear is not fixed. To address this, we adopt a strategy from prior work (Chen et al., 2022; Gao

et al., 2024), which minimizes the likelihood of generating termination tokens across all positions in the output sequence:

$$\mathcal{L}_{\text{DT}} = \frac{1}{|y|} \sum_{t=1}^{|y|} \Big[ p(y_t = \text{EOS} \mid \{x, x'\}, y_{<t}) + p(y_t = \text{EOT} \mid \{x, x'\}, y_{<t}) \Big].$$

This objective discourages premature termination, encouraging the model to continue generating extended reasoning and answer conclusions before finalizing its output.

## 2.4 Optimization

Optimizing suffixes in the text domain presents a unique challenge due to the discrete nature of language. Unlike continuous domains (e.g., images), where gradients can be directly applied to pixel values, LMs operate on sequences of discrete tokens drawn from a fixed vocabulary. As a result, standard gradient-based optimization cannot be directly applied to manipulate individual tokens.

To address this, we adopt the Greedy Coordinate Gradient-based Search (GCG) framework (Zou et al., 2023), which has demonstrated strong performance in adversarial text generation. GCG linearizes the loss landscape by computing gradients with respect to input embeddings and identifying substitutions that are most likely to improve the loss. Specifically, for a given token position $i$ in the suffix, we compute the gradient of the loss with respect to its embedding and search for the token $x'_i$ that maximally improves the objective. Formally:

$$x'_i = \arg\max_{w \in V} \left\langle \nabla_{e(x_i)} \mathcal{L},\ e(w) - e(x_i) \right\rangle, \tag{5}$$

where $\nabla_{e(x_i)} \mathcal{L}$ denotes the gradient of the loss with respect to the embedding of token $x_i$, and $e(w)$ is the embedding of candidate token $w$. This inner product quantifies the expected gain from substituting $x_i$ with $w$, and the best candidate is selected greedily. Our overall objective combines the three loss components introduced previously:

$$\mathcal{L}_{long} = \alpha \cdot \mathcal{L}_{\text{PCE}} + \beta \cdot \mathcal{L}_{\text{ER}} + \gamma \cdot \mathcal{L}_{\text{DT}}. \tag{6}$$

In this work, we employ a fixed-length suffix-based optimization, where a set of tokens is appended to the end of the original prompt. Each token in the suffix is iteratively updated using GCG to minimize the combined loss. Although this paper focuses on crafting adversarial suffixes, it is important to note that our approach is *method-agnostic* and can be adapted to any paradigms. For instance, alternative strategies such as character-level perturbations (e.g., typos) can be incorporated, as shown in prior work (Gan et al., 2024). This flexible framework facilitates the efficient generation of adversarial inputs tailored to different attack constraints.

## 3 Experiments

### 3.1 Experimental Setups

**Models and Datasets.** We optimize adversarial suffixes and evaluate them on two reasoning LLMs: DeepSeek-R1-distill-LLaMA-8B and DeepSeek-R1-distill-Qwen-7B.[2] Both models are distilled variants of DeepSeek-R1. We report results under two decoding strategies: greedy decoding and sampling decoding. For sampling, we set the temperature to 0.6, apply nucleus sampling with top-$p = 0.95$, and repeat 10 times. To assess cross-model transferability, we additionally evaluate the attack on larger-scale models, including o3-mini, GPT-OSS (with low and medium reasoning), DeepSeek-R1, and QWQ-32B, using their respective default decoding settings. Our evaluation is conducted on two widely used mathematical reasoning benchmarks: GSM8K (Cobbe et al., 2021) and ORCA (Mitra et al., 2024). For each dataset, we randomly sample 50 examples for both optimization and evaluation. More details are provided in App. C.4.

**Attack Setup.** For DSPy, we use the COPRO optimizer with "gpt-4o-mini" to refine the CoT prompt. Specifically, we use 10 training examples from the GSM8K dataset for DSPy optimization

---

[2]For simplicity, we omit the prefix DeepSeek-R1-distill throughout the remainder of the paper.

| Models | Methods | GSM8K | | | | | | ORCA | | | | | |
| | | Rea | Ans | Full | Lat | Ent | Acc | Rea | Ans | Full | Lat | Ent | Acc |
|---|---|---|---|---|---|---|---|---|---|---|---|---|---|
| LLaMA | Original | 668 | 239 | 907 | 24.3 | 4628 | 72% | 499 | 213 | 712 | 19.3 | 4366 | **82%** |
| | Random | 447 | 264 | 711 | 19.0 | 3571 | 74% | 440 | 228 | 668 | 17.6 | 3570 | 76% |
| | CoT | 574 | **266** | 839 | 22.2 | 4712 | 70% | 344 | **259** | 603 | 14.7 | 2789 | **82%** |
| | Engorgio | 168 | 274 | 443 | 10.4 | 1804 | 68% | 189 | 219 | 408 | 9.19 | 1644 | 82% |
| | CatAttack | 496 | 232 | 729 | 19.7 | 3959 | 76% | 338 | 233 | 571 | 15.3 | 3013 | 78% |
| | Ours | **1914** | 160 | **2074** | **54.9** | **12827** | **92%** | **1575** | 167 | **1743** | **47.2** | **9929** | 80% |
| Qwen | Original | 237 | 282 | 519 | 12.8 | 2498 | 84% | 531 | 220 | 750 | 18.4 | 4034 | 84% |
| | Random | 159 | 294 | 453 | 11.2 | 2097 | 82% | 527 | 225 | 752 | 18.7 | 3505 | 86% |
| | CoT | 169 | **310** | 479 | 11.9 | 2535 | 82% | 379 | **248** | 626 | 15.6 | 2910 | 86% |
| | Engorgio | 832 | 274 | 1106 | 24.4 | 4661 | 82% | 273 | 203 | 476 | 9.99 | 1811 | 86% |
| | CatAttack | 167 | 295 | 461 | 11.3 | 2171 | 78% | 532 | 234 | 766 | 18.5 | 4040 | 82% |
| | Ours | **1531** | 193 | **1724** | **42.4** | **8188** | **88%** | **1459** | 166 | **1624** | **39.6** | **9155** | **88%** |

Table 1: The token length for reasoning (Rea), answer (Ans), and full output (Full); inference latency (Lat, in seconds); energy consumption (Ene, in joules); and task accuracy (Acc). Experimental results across methods under greedy decoding. **Bold** indicates the best result.

and evaluate on a separate set of 10 randomly selected test samples. Due to computational constraints, we restrict target outputs to a maximum length of 3,000 tokens. For the PCE Loss, we set the token selection threshold $K = 1$, and for the ER Loss, we use $n = 5$. The overall loss function combines the three components using the following weighting coefficients: $\alpha = 1$, $\beta = 50$, and $\gamma = 1$. We fix the length of the adversarial suffix to 10 tokens. During optimization, we apply the GCG algorithm for 1,000 steps per input. The candidate pool size is set to 64, and at each step, the top 64 candidate tokens are retained.

**Evaluation Metrics.** To evaluate the effectiveness of our adversarial attack, we consider three primary metrics: (1) output sequence length (tokens), (2) inference latency (seconds), and (3) energy consumption (Joules). Energy usage is measured using the NVIDIA Management Library, following the methodology introduced by (Shumailov et al., 2021). To ensure consistency and fair comparison, all inference is performed using the HuggingFace pipeline (Wolf et al., 2019) on a single hardware (NVIDIA A100 80GB). Each inference is repeated three times to reduce the impact of runtime variability. To assess model utility, we extract final answers from the generated outputs using "Meta-Llama-3.1-8B-Instruct", and compute accuracy by comparing the extracted answers against ground-truth labels. The exact prompt used for extraction is provided in App. C.1.

**Baselines.** We compare our attack against several baseline methods:

- **Random**: 10 randomly sampled tokens.

- **Standard CoT (Wei et al., 2022)**: A widely used CoT prompt that appends the phrase "Let's think step by step."

- **Engorgio (Dong et al., 2025)**: A method for crafting "inference-cost" attacks via adversarial prompts that force auto-regressive LLMs to produce excessively long outputs.

- **CatAttack (Rajeev et al., 2025)**: A prompt-based adversarial strategy that appends the distractor statement "Interesting fact: cats sleep most of their lives," which has been shown to induce incorrect reasoning outputs.

## 3.2 MAIN RESULTS

**Performance.** As shown in Table 1 and Table 12, our adversarial suffix substantially increases computational overhead while preserving task accuracy across all settings. For example, our attack causes LLaMA to generate significantly longer outputs on the GSM8K dataset with greedy decoding, increasing the average reasoning length by 3x from 668 to 1,914 tokens. This is accompanied by a corresponding increase in energy consumption (from 4,628J to 12,827J) and latency (from 24.3s to 54.9s). A similar trend is observed for Qwen, where the average reasoning length increases by 6.5x, demonstrating the effectiveness of our attack across different model architectures. Under sampling-based decoding, the attack remains robust. The reasoning length increases by 3x for LLaMA and 4x for Qwen on GSM8K, with similar results observed on the ORCA dataset.

| Models | Methods | GSM8K | | | | | | ORCA | | | | | |
|--------|---------|-----|-----|------|------|-------|-----|------|-----|------|-------|-------|-----|
| | | Rea | Ans | Full | Lat | Ent | Acc | Rea | Ans | Full | Lat | Ent | Acc |
| LLaMA | Original | 556 | 257 | 812 | 71.5 | 9778 | 76% | 550 | 248 | 798 | 86.7 | 10490 | 81% |
| | Random | 401 | 270 | 671 | 54.0 | 7928 | 72% | 493 | 244 | 737 | 80.6 | 9689 | 81% |
| | CoT | 476 | **280** | 757 | 68.5 | 7713 | 75% | 402 | **266** | 668 | 75.9 | 9148 | 80% |
| | Engorgio | 161 | 282 | 443 | 15.3 | 3055 | 68% | 249 | 234 | 484 | 39.1 | 9663 | 80% |
| | CatAttack | 528 | 257 | 785 | 64.8 | 9818 | 77% | 475 | 224 | 700 | 81.4 | 10680 | 82% |
| | Ours | **1437** | 204 | **1641** | **197.0** | **21228** | **90%** | **1425** | 206 | **1631** | **178.2** | **19243** | **87%** |
| Qwen | Original | 345 | 277 | 622 | 37.0 | 5996 | 82% | 452 | 250 | 701 | 51.2 | 7538 | 84% |
| | Random | 221 | 296 | 518 | 23.4 | 4245 | 84% | 501 | 254 | 755 | 54.5 | 7353 | 86% |
| | CoT | 176 | **308** | 484 | 16.0 | 3799 | 85% | 293 | **274** | 567 | 33.1 | 5722 | **87%** |
| | Engorgio | 191 | 300 | 490 | 20.5 | 4352 | 81% | 268 | 205 | 473 | 21.9 | 4718 | 78% |
| | CatAttack | 187 | 295 | 482 | 17.4 | 3398 | 81% | 432 | 253 | 686 | 50.0 | 6458 | 83% |
| | Ours | **1479** | 217 | **1696** | **149.1** | **16031** | **91%** | **1238** | 183 | **1421** | **111.2** | **14747** | **87%** |

Table 2: Experimental results across methods under sampling decoding.

In comparison, baseline methods generally induce relatively short reasoning. For example, CoT prompts produce shorter outputs than our adversarial prompt on LLaMA for GSM8K under greedy decoding (839 vs. 2,074 tokens), indicating the limitations of standard methods in eliciting excessive reasoning behavior. More broadly, our results suggest that reasoning LLMs are resistant to short prompting, as neither CoT nor CatAttack reliably triggers long reasoning. Interestingly, we observe a consistent inverse correlation between the lengths of reasoning and answer segments. We hypothesize that as the model allocates more capacity to the reasoning phase, the corresponding answer portion becomes shorter. Also, the increase in reasoning length does not degrade task accuracy. In most cases, our triggered excessive reasoning produces meaningless reasoning, as shown in App. C.2 (sample 1). However, we also find that in a few cases, the regular reasoning is too simple and fails to solve the problem, while our triggered excessive reasoning occasionally fixes it, as shown in App. C.2 (sample 2). For example, the original response contains a single solution (reasoning path) and it actually leads to an incorrect calculation. In contrast, our attack triggers excessive reasoning that explicitly extends the thinking with multiple solutions; however, we can see that the last solution (reasoning path) is redundant since the answer has already been inferred correctly. In general, the majority of problems can be solved with baseline prompting, and our attack, in most cases, simply extends the reasoning unnecessarily, leading to computational waste. Together, these results demonstrate that our attack amplifies the computational burden of reasoning models without compromising their effectiveness. We also provide an additional study on ELI5, optimized suffix, and sample outputs in App. C.2.

**Analysis.** To determine whether our adversarial suffixes truly elicit excessive reasoning rather than merely increasing output length, we conduct an analysis of the generated outputs. First, we observe a substantial increase in the average number of reasoning sentences. For LLaMA, the average rises from 32 to 88, and for Qwen, from 11 to 74, when comparing outputs generated from original prompts to those generated with adversarial suffixes. This indicates that our attack expands the number of reasoning rather than just inflating output length.

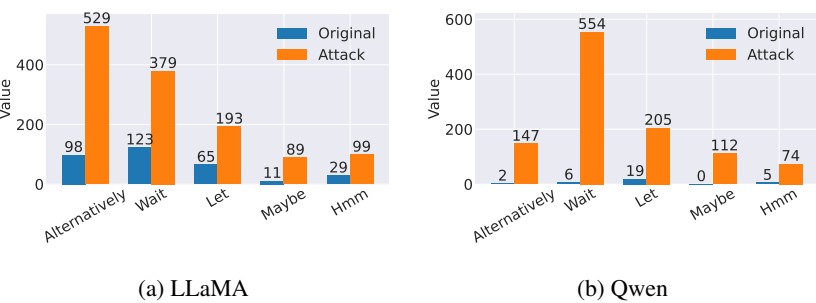

(a) LLaMA             (b) Qwen

Figure 2: Token counts between the generated outputs from the original and adversarial prompts.

Next, we examine the distribution of the first two tokens in reasoning sentences (Fig. 2). The results reveal distinct lexical patterns between the two models. LLaMA often begins with deliberative tokens such as *"Alternatively"* and *"Wait,"* associated with recursive reasoning, while Qwen is less

| Models | Methods | GSM8K | | | ORCA | | |
|--------|---------|-------|-----|----------|------|-----|----------|
| | | Toks | Acc | Toks/Acc | Toks | Acc | Toks/Acc |
| LLaMA | Original | 907 | 72% | 12.6 | 712 | **82%** | 8.7 |
| | Ours | **2074** | **92%** | **22.5 (1.8×)** | **1743** | 80% | **21.8 (2.5×)** |
| Qwen | Original | 519 | 84% | 6.2 | 750 | 84% | 8.9 |
| | Ours | **1724** | **88%** | **19.6 (3.2×)** | **1624** | **88%** | **18.5 (2.1×)** |

Table 3: Tokens per accuracy (Toks/Acc) under Original vs. our attack with greedy decoding. Ratios show relative increases over Original.

| Model | LLaMA | | | | | Qwen | | | | |
|-------|-------|--------|------|----------|-------|--------|--------|------|----------|-------|
| | Reason | Answer | Full | Accuracy | Match | Reason | Answer | Full | Accuracy | Match |
| QWQ | 1761 (-151) | 231 (-6) | 1992 (-157) | 93% (-3%) | 62% | 2489 (+577) | 317 (+80) | 2806 (+657) | 98% (+2%) | 100% |
| R1 | 997 (-74) | 185 (-13) | 1182 (-87) | 95% (-2%) | 4% | 1295 (+224) | 233 (+36) | 1528 (+260) | 93% (-4%) | 60% |
| o3-mini | 446 (+199) | 184 (+46) | 630 (+245) | 90% (+1%) | 20% | 645 (+398) | 336 (+199) | 982 (+596) | 89% (0%) | 70% |
| gpt5-mini | 343 (+175) | 106 (+34) | 449 (+209) | 77% (-7%) | 60% | 578 (+410) | 210 (+138) | 787 (+547) | 83% (-2%) | 70% |
| Gemini-Flash | 665 (+146) | 350 (+76) | 1015 (+222) | 86% (-3%) | – | 874 (+355) | 513 (+239) | 1388 (+594) | 90% (+1%) | – |
| OSS (low) | 79 (+13) | 170 (+35) | 249 (+48) | 92% (+2%) | 20% | 95 (+19) | 316 (+167) | 411 (+186) | 90% (0%) | 70% |
| OSS (medium) | 569 (+149) | 237 (+111) | 806 (+259) | 90% (+1%) | 20% | 3015 (+2569) | 348 (+206) | 3363 (+2775) | 88% (-2%) | 70% |

Table 4: Transferability analysis of adversarial suffixes originally optimized for LLaMA and Qwen.

sensitive to *"Alternatively,"* suggesting that the expression of excessive reasoning may manifest differently across architectures. Notably, Qwen does not exhibit the same degree of excessive reasoning as LLaMA under standard conditions but becomes vulnerable when adversarial suffixes are applied. Tokens such as *"Let"*, *"Maybe"*, and *"Hmm"*, which are difficult to detect manually, highlight the utility of ER Loss combined with automated token selection in surfacing subtle prompts that trigger excessive reasoning behavior.

Finally, we use the Tokens per Accuracy metric to measure reasoning efficiency in Table 3. Lower values indicate concise, effective reasoning, while higher values reflect inefficient reasoning. Across all model–dataset pairs, adversarial suffixes inflate this metric. For instance, on GSM8K, LLaMA's tokens per accuracy point increase from 12.6 to 22.5 (a 1.8x increase), indicating that the model requires substantially more tokens per correct answer. This consistent pattern demonstrates that our method successfully induces inefficient, overextended reasoning.

**Transferability.** We evaluate the transferability of our adversarial suffixes to larger commercial LMs, including o3-mini, GPT-OSS, DeepSeek-R1, and QWQ. Specifically, we test adversarial suffixes optimized on the LLaMA and Qwen models for the GSM8K dataset, with results summarized in Table 4. Our findings show that these adversarial suffixes generalize effectively, consistently promoting longer output sequences without degrading task accuracy. For the OpenAI model family, both LLaMA- and Qwen-optimized suffixes successfully increase output length. For example, suffixes optimized on LLaMA lead to a 245-token increase in total output length for o3-mini, and Qwen-optimized suffixes yield a 596-token increase.

In contrast, transferability to DeepSeek-R1 appears to depend on the source model. Qwen-optimized suffixes result in a 260-token increase, whereas LLaMA-optimized suffixes fail to induce longer outputs. We hypothesize that this discrepancy is due to the token mismatch between the source and target models. We report the Match metric to reflect the token alignment rate between the source model and the target model. Specifically, we check whether the token length of the optimized suffix—when encoded by the source model and by the target model (e.g., gpt5)—is the same. If the lengths match, it indicates that none of the tokens in the optimized suffix are split or merged when processed by the target model. We observe that when the Match score is high, the overall length is also high, demonstrating a positive relationship between the two. These results suggest that while architectural differences influence the degree of computational overhead, token alignment plays a critical role in the transferability of adversarial prompts.

## 3.3 ABLATION STUDIES

We conduct a series of ablation studies to assess the impact of different experimental configurations, including the introduction of the PCE Loss and the individual contribution of each loss component.

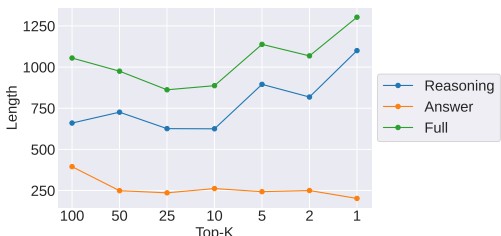

Figure 3: Impact of varying the top-K most informative tokens on LLaMA under greedy decoding.

| Setup | Reason | Answer | Full | Latency | Energy | Accuracy |
|---|---|---|---|---|---|---|
| $\mathcal{L}_{PCE}$ | 1100 | **202** | 1303 | 35.0 | 7016 | 84% |
| $\mathcal{L}_{PCE} + \mathcal{L}_{ER}$ | 1169 | 188 | 1357 | 36.1 | 8089 | 88% |
| $\mathcal{L}_{PCE} + \mathcal{L}_{DT}$ | 1447 | 201 | 1648 | 43.9 | 9558 | 88% |
| $\mathcal{L}_{PCE} + \mathcal{L}_{ER} + \mathcal{L}_{DT}$ | **1914** | 160 | **2074** | **54.9** | **12827** | **92%** |

Table 5: Different loss combinations on LLaMA under greedy decoding.

**PCE Loss.** We begin by evaluating the effectiveness of the proposed PCE Loss by varying the proportion of top-$K$ tokens from 100% (Original CE) to 1%, as shown in Fig. 3. The results show that focusing optimization on the top 5%, and particularly the top 1% of tokens, consistently outperforms applying loss uniformly across all tokens. Peak performance is observed when focusing solely on the top 1%, with the number of reasoning tokens increasing from 660 to 1,100. This pattern suggests that selectively emphasizing a small subset of salient, prompt-dependent tokens can more effectively induce extended reasoning behavior. Additionally, we observe an inverse relationship between the number of reasoning and answer tokens, implying a redistribution of the model's generative capacity toward reasoning content. These findings underscore the value of targeted token optimization and demonstrate that prioritizing high-impact tokens is more effective than uniformly distributing the loss across the entire sequence.

**Loss Objectives.** Second, we evaluate the individual contributions of each loss function and assess their collective impact as presented in Table 5. The results show that optimizing each loss independently leads to an increase in output sequence length, and the combination of all three losses yields the most substantial gains in both sequence length and computational burden. For instance, the full composite loss achieves the longest average output (up to 2,074 tokens), the highest inference latency (54.9 seconds), and the greatest energy consumption (12,827J). These results underscore the synergistic effect of combining all three objectives.

To further analyze the behavior encouraged by the ER Loss, we visualize a word cloud of the most frequently prioritized tokens in Fig. 5. Common deliberative tokens identified in prior work, such as *"Alternatively"* and *"Wait"*, are prominently featured. In addition, our method surfaces previously underexplored tokens such as *"Maybe"* and *"Hmm"*, which act as effective triggers for extended reasoning. These findings confirm that the joint loss formulation effectively amplifies reasoning behavior while preserving task accuracy, and that the ER Loss successfully uncovers subtle lexical cues indicative of recursive reasoning.

## 4 POTENTIAL DEFENSES

Currently, no defenses are proposed explicitly against reasoning-based attacks. To evaluate potential countermeasures, we test four strategies: two at the input level and two at the output level, with details summarized in App.B. Naïve methods, such as perplexity-based filtering, yield high false positives and significantly degrade user experience. Safety classifiers are another possible input-level defense, but our attack bypasses them with high success rates. At the output level, we examine two decoding techniques aimed at curbing excessive reasoning. Thought Switching Penalty (TIP) (Wang et al., 2025) penalizes reasoning-disruptive tokens but fails to shorten overall outputs, while Dynamic Early Exit for Reasoning (DEER) (Yang et al., 2025) halts decoding once a confident

answer is reached but may even reduce model utility. Overall, these findings show that our attack remains robust against different defenses.

## 5 RELATED WORK

**Adversarial Attacks on LLMs.** Adversarial attacks on LLMs present unique challenges due to the discrete nature of text, which limits the applicability of gradient-based techniques commonly used in vision tasks. Early work, such as HotFlip (Ebrahimi et al., 2018), has introduced token-level perturbations as vectors, enabling efficient gradient-guided attacks on classification models. This approach was further developed by Universal Adversarial Triggers (UAT) (Wallace et al., 2019), which identified fixed token sequences that could consistently manipulate model outputs across diverse input data. More recently, GCG (Zou et al., 2023) demonstrated strong performance in generating adversarial suffixes that elicit specific responses. BEAST (Sadasivan et al., 2024) improves upon GCG by incorporating beam search to enhance linguistic coherence, thereby increasing the effectiveness of adversarial prompts in bypassing safety filters.

**Denial-of-Service Attacks on LLMs.** Denial-of-service (DoS) attacks on LLMs aim to degrade system performance by manipulating input prompts to increase computational cost, latency, or energy consumption. For example, Sponge attacks (Shumailov et al., 2021) maximize the L2 norm of internal activations, thereby exhausting system resources during inference. Subsequent work on energy-latency manipulation introduced crafted inputs that prolong decoding time or increase memory usage. NICGSlowDown (Chen et al., 2022) demonstrated that minimizing the probability of EOS tokens results in excessively long outputs in captioning models, significantly increasing the computational load. More recently, Overthink (Kumar et al., 2025) was proposed as an indirect prompt injection method that utilizes decoy tasks to induce unnecessary reasoning overhead. In contrast, Crabs (Zhang et al., 2025) shares a similar idea but places the decoy within the user query. Additionally, Engorgio Prompt (Dong et al., 2025) demonstrates how attackers can trigger excessive resource usage in white-box settings with a re-parameterization design.

## 6 CONCLUSION

In this work, we present a novel adversarial attack on reasoning LLMs that induces computational overhead during inference. Our approach appends adversarial suffixes that trigger extended reasoning trajectories, optimized through a composite loss function that maximizes output length and complexity. Empirical results demonstrate that the method reliably increases sequence length, inference latency, and energy consumption. Moreover, the strong cross-model transferability highlights the practical relevance of this threat.

**Limitations.** Based on the analysis, we can see that the choice of optimized tokens is also critical and constrained by the tokenizer used. Right now, many commercial APIs—such as OpenAI—release their tokenizers, and tokenizer inference attacks also exist and can be used to recover unknown tokenizers. In future work, the selection of optimized tokens can be incorporated as an additional optimization constraint, not only for this attack but for other adversarial methods as well.

## ETHICS STATEMENT

This study utilizes publicly available datasets and models, with appropriate citations provided. No private or sensitive information is used, and the work does not include any harmful content. Because the research relies exclusively on publicly available data and does not involve human participants, it does not qualify as human subjects research under Institutional Review Board (IRB) definitions. Accordingly, we conclude that this paper raises no ethical concerns.

## REPRODUCIBILITY STATEMENT

We provide detailed descriptions of the training and evaluation procedures used in our experiments. In particular, we describe our proposed attack in detail (see Sec. 2), followed by the specifications

of the models, datasets, and decoding strategies used (see Sec. 3.1). All prompting templates are included in the Appendix for reference. The code will be released upon publication of this paper.

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

## A  THE USE OF LARGE LANGUAGE MODELS

In this paper, LLMs are employed to enhance the clarity, fluency, and overall quality of writing. Their use is limited to language refinement and does not extend to the design, execution, or analysis of the experiments. Also, all polished texts are double-checked by authors to ensure accuracy, avoid overclaims, and prevent confusion.

## B  DEFENSE STUDIES

**Input-level.** At the input level, we begin by evaluating a perplexity-based filtering strategy, following the similar setup described in (Dong et al., 2025). We compute universal perplexity thresholds using the Open-Platypus dataset, assuming the model owner has no prior knowledge of the adversarial suffix. Specifically, we select the 25th, 50th, and 75th percentile quantiles as thresholds to filter prompts exhibiting unusually high perplexity.

We test the filter on both clean prompts and prompts appended with adversarial suffixes as shown in Table 7. While this method achieves a 100% true positive rate in detecting adversarial prompts, it also results in high false positive rates for clean prompts—100%, 78%, and 16% on LLaMA, and 100%, 84%, and 28% on Qwen at the respective thresholds. These results suggest that our optimized suffixes are difficult to identify when the model owner lacks prior knowledge of their

| | LLaMA | | | | Qwen | | | |
|---|---|---|---|---|---|---|---|---|
| | Reason | Answer | Full | Accuracy | Reason | Answer | Full | Accuracy |
| Base | 668 | 239 | 907 | 72% | 237 | 282 | 519 | 84% |
| Base + TIP | 422 | 250 | 673 | 74% | 264 | 283 | 547 | 84% |
| Ours | 1914 | 160 | 2074 | 92% | 1531 | 193 | 1724 | 88% |
| Ours + TIP | 1863 | 173 | 2037 | 94% | 1464 | 198 | 1661 | 84% |
| Base | 668 | 239 | 907 | 72% | 237 | 282 | 519 | 84% |
| Base + DEER | 247 | 237 | 484 | 70% | 217 | 268 | 485 | 86% |
| Ours | 1914 | 160 | 2074 | 92% | 1531 | 193 | 1724 | 88% |
| Ours + DEER | 1074 | 216 | 1290 | 72% | 426 | 518 | 945 | 88% |

Table 6: Output-level defense results for GSM8k on LLaMA and Qwen.

| Quantile | LLaMA | | Qwen | |
|---|---|---|---|---|
| | FP (%) | TP (%) | FP (%) | TP (%) |
| 25 | 100 | 100 | 100 | 100 |
| 50 | 78 | 100 | 84 | 100 |
| 75 | 16 | 100 | 28 | 100 |

Table 7: False positive rate (FP) for filtering clean prompts and true positive rate (TP) for filtering attack prompts using perplexity-based filtering across different quantiles.

structure. Also, applying high-percentile quantile thresholds may severely degrade the utility of the model on benign inputs. For the second defense, we apply an open-source safety classifier, "meta-llama/Llama-Prompt-Guard-2-86M," to flag unsafe inputs. The classifier detects only 8% of adversarial prompts on LLaMA and 6% on Qwen. These findings indicate that our adversarial suffixes are able to bypass the classifier in most cases.

**Output-level.** At the output level, we evaluate two complementary mitigation techniques. The first, Thought Switching Penalty (TIP) (Wang et al., 2025), penalizes the generation of reasoning-disruptive tokens such as "Wait" or "Alternatively." The second, Dynamic Early Exit for Reasoning (DEER) (Yang et al., 2025), reduces unnecessary token generation by terminating decoding early once a confident answer is detected.

As shown in Table 6, on LLaMA, our attack generated 1,914 reasoning tokens and 160 answer tokens, totaling 2,074 tokens, with an accuracy of 92%. Applying TIP led to a modest reduction in total tokens (2,037) and a slight increase in accuracy to 94%. In contrast, DEER reduced the length of reasoning (to 1,290 tokens) but caused a more substantial performance drop, lowering accuracy to 72%. A similar trend was observed on Qwen: TIP preserved accuracy while modestly reducing output length, whereas DEER can only reduce the total token count (945 tokens). In addition, DEER requires extra processing time, which also incurs additional latency and energy costs. Overall, these results suggest that TIP provides only limited mitigation against our attack, while DEER is more effective at curbing excessive reasoning, but at the cost of degraded task performance and additional computational resources. Nevertheless, our attack still incurs substantial resource overhead due to excessive and redundant token generation compared to Original.

## C  ADDITIONAL EXPERIMENTAL DETAILS

In this section, we present additional details of our experimental setup, including the prompting templates, sample outputs and suffixes, outputs and prompts from DSPy, analysis of the attack behavior, descriptions of the models and datasets employed, as well as the computational resources utilized.

## C.1 PROMPT TEMPLATE

---
**Prompting template for extracting answer**

Here is a math question and a model's answer about this question.
Please extract the EXACT number from the answer text as the final answer for question.
QUESTION: {}
ANSWER: {}
Final format should be a legal 'number' without any suffix such as '$'.
The final answer is:

---

## C.2 SAMPLES

---
**Prompt + Optimized Suffix**

Janet's ducks lay 16 eggs per day. She eats three for breakfast every morning and bakes muffins for her friends every day with four. She sells the remainder at the farmers' market daily for $2 per fresh duck egg. How much in dollars does she make every day at the farmers' market? **?! Contains more tricks........ Elijahnumeric paradox nearer[/**

---

---
**Sample response 1 from our attack**

Okay, so I need to figure out how much money Janet makes every day from selling her duck eggs. Let me start by breaking down the information given.
First, Janet's ducks lay 16 eggs per day. That's the total number of eggs she gets each day. Now, she has some daily responsibilities that involve using these eggs. She eats three for breakfast every morning, so that's 3 eggs used daily. Then, she bakes muffins for her friends every day with four eggs. Wait, does that mean she uses 4 eggs per day ...
So far, she's using 3 eggs for breakfast and 4 for baking, which totals 7 eggs per day. She lays 16 eggs ...
So, 9 eggs sold at $2 each would be 9 times 2, which equals $18. Therefore, Janet makes $18 every day from selling her eggs ...
Wait, let me double-check to make sure I didn't miss anything. She lays 16, uses 3 for breakfast, 4 for baking, so 16 - 3 - 4 = 9. Yes, that's correct. And each egg is sold for $2, so 9 times 2 is indeed $18. I think that's all there is to it. ... ...

---

---
**Sample response 1 from original**

First, determine the total number of eggs Janet's ducks lay each day, which is 16.
Next, calculate the number of eggs Janet uses for breakfast and baking muffins. She eats 3 eggs for breakfast and uses 4 eggs for baking, totaling 7 eggs.
Subtract the used eggs from the total to find out how many eggs she has left to sell: 16 - 7 = 9 eggs.
Finally, multiply the number of eggs she sells by the price per egg to find out how much money she makes daily: 9 eggs * $2 per egg = $18 ...

---

---
**Sample question 2**

Raymond and Samantha are cousins.
Raymond was born 6 years before Samantha.

---

Raymond had a son at the age of 23.
If Samantha is now 31, how many years ago was Raymond's son born?

**Sample response 2 from our attack**

Okay, so I'm trying to figure out this problem about Raymond and Samantha ...
Raymond and Samantha are cousins. Raymond was born 6 years before Samantha. Raymond had a son at the age of 23 ...
Alright, so Raymond was born 6 years before Samantha ...
Wait, no, that's not right. If Raymond was born 6 years before Samantha ...
Yeah, that makes sense because if you subtract 6 years from Samantha's age ...
So, 31 - 6 = 25?
Wait, no, that's the opposite ...
Raymond had a son at the age of 23.
So, Raymond was 23 when his son was born.
Now, Raymond is 37, so the time between Raymond having his son and now is 37 - 23 = 14 years.
So, the son was born 14 years ago.
Wait, but let me double-check that. If Raymond was 23 when his son was born, and he is now 37, then 37 - 23 = 14 years have passed since the son was born.
So, the son is now 14 years old, which means he was born 14 years ago. That seems right.
...
</think>... **Answer:** Raymond's son was born 14 years ago.

**Sample response 2 from original**

First, I need to determine Samantha's age when Raymond's son was born.
Since Samantha is currently 31 years old, I'll subtract 6 years from her current age to find out how old she was when Raymond was born.
This gives me 25 years old.
Next, I'll figure out how long ago Raymond's son was born.
Raymond had his son at the age of 23.
To find out how many years ago this happened, I'll subtract 23 from Samantha's age at that time.
So, 25 minus 23 equals 2 years ago.
Therefore, Raymond's son was born 2 years ago.
</think>... Final Answer:\n\\[\n \\boxed{\\{ 2}}\n\\]

## C.3 DSPY

|          | Mean |
|----------|------|
| Raw      | 805  |
| CoT      | 717  |
| CoT+DSPy | 1247 |

Table 8: The statistics of sample outputs with different promptings.

**DSPy-optimized prompt**

Provide a detailed breakdown of the problem or topic at hand, considering every aspect and nuance. Start from an introductory overview, followed by dissecting the components logically, integrating examples, addressing possible counterarguments or misconceptions, and uniformly

elaborating on each point with explanatory details. Prioritize presenting a thorough, analytical approach that can include speculative insights, hypothetical scenarios, or alternative viewpoints, without rushing to conclusions. Aim to showcase a clear conceptualization followed by logical reasoning, actively prompting self-reflection on the implications of your insights.

---

**Sample output from DSPy-optimized prompt**

Okay, so I need to figure out how much money Janet makes every day from selling her duck eggs at the farmers' market. Let me start by breaking down the problem step by step.
First, Janet's ducks lay 16 eggs per day. That's the total number of eggs she gets each day. Now, she has some daily expenses and uses related to these eggs. She eats three eggs for breakfast every morning. So that's 3 eggs used for breakfast. Then, she bakes muffins for her friends every day, using four eggs each time. **Wait**, does she bake muffins once a day using four eggs, or does she bake them multiple times? The problem says "bakes muffins for her friends every day with four," which is a bit ambiguous. I think it probably means she uses four eggs per day for baking muffins. So that's another four eggs.
So, adding up her daily egg usage: 3 eggs for breakfast plus 4 eggs for muffins equals 7 eggs used each day. Now, she lays 16 eggs, so how many does she have left after using 7? Let me subtract: 16 minus 7 equals 9 eggs remaining. These 9 eggs are sold at the farmers' market, and she sells each for $2. So, 9 eggs times $2 per egg would give her $18 per day.
**Wait**, let me double-check that. 16 eggs laid minus 7 used equals 9 sold. 9 times 2 is indeed 18.
...

---

## C.4 EXPERIMENTAL SETUPS

All models in our experiments are downloaded from HuggingFace. DeepSeek-R1-Distill-Llama-8B and DeepSeek-R1-Distill-Qwen-7B are originally licensed under the Apache 2.0 License. The GSM8K and ORCA datasets are under the MIT License. All experiments in the paper were conducted on an A100 (80GB) compute node. Also, we interact with o3-mini via the OpenAI API, and with DeepSeek-R1 and QWQ-32B via the Baidu Cloud API, to simulate real-world deployment conditions.

## C.5 TARGET OUTPUT CONSTRUCTION

| Setup | Reason | Answer | Full | Latency | Energy | Accuracy |
|---|---|---|---|---|---|---|
| Raw | 1695 | 190 | 1885 | 50.9 | 12120 | 80% |
| CoT | 1060 | **224** | 1283 | 34.0 | 7719 | 80% |
| CoT + DSPy | **1914** | 160 | **2074** | **54.9** | **12827** | **92%** |

Table 9: Different target constructions on LLaMA under greedy decoding.

| Setup | Reason | Answer | Full | Latency | Energy | Accuracy |
|---|---|---|---|---|---|---|
| Raw | 1396 | 223 | 1619 | 206.8 | **20252** | 82% |
| CoT | 1034 | **250** | 1283 | 150.2 | 15685 | 79% |
| CoT + DSPy | **1501** | 216 | **1718** | **209.9** | 19491 | **87%** |

Table 10: Different target construction strategies on LLaMA under sampling decoding.

We evaluate several strategies for constructing target outputs to guide adversarial optimization, as summarized in Table 9 and 10. The comparison includes a raw baseline (no additional prompt), a standard CoT prompt, and a DSPy-optimized CoT prompt. Interestingly, we find that the standard CoT prompt does not consistently produce longer reasoning sequences; in some cases, it even results in shorter outputs than raw prompting, highlighting its limitations in eliciting extended reasoning.

In contrast, DSPy-optimized CoT prompts increase the average output length from 1,283 to 2,074 tokens under greedy decoding compared to CoT prompts, with corresponding increases in both energy consumption and task accuracy. These results highlight the critical role of target output quality in guiding adversarial optimization. Longer reasoning sequences, especially those produced via DSPy, serve as more effective targets for inducing excessive computation. This reinforces the importance of target construction in maximizing the efficacy of our attack.

## C.6 ADDITIONAL RESULTS

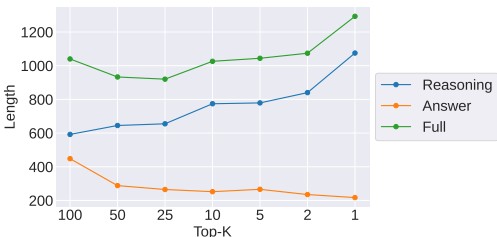

Figure 4: Impact of varying the top-K most informative tokens on LLaMA under sampling decoding.

| Setup | Reason | Answer | Full | Latency | Energy | Accuracy |
|---|---|---|---|---|---|---|
| $\mathcal{L}_{PCE}$ | 1104 | 218 | 1322 | 143.9 | 19810 | 86% |
| $\mathcal{L}_{PCE} + \mathcal{L}_{ER}$ | 1095 | 205 | 1301 | 124.9 | 14879 | 87% |
| $\mathcal{L}_{PCE} + \mathcal{L}_{DT}$ | 1184 | **223** | 1407 | 145.3 | 17731 | 88% |
| $\mathcal{L}_{PCE} + \mathcal{L}_{ER} + \mathcal{L}_{DT}$ | **1437** | 204 | **1641** | **197.0** | **21228** | **90%** |

Table 11: Ablation study of loss objectives combinations on LLaMA under sampling decoding.

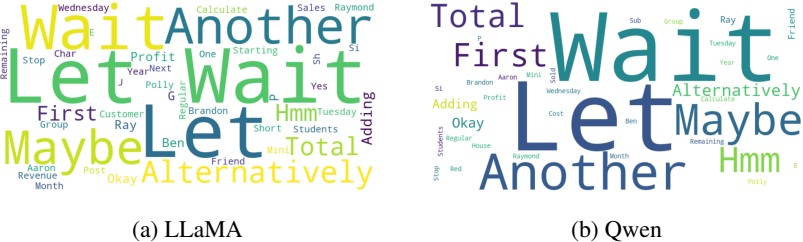

(a) LLaMA           (b) Qwen

Figure 5: Tokens used for ER Loss. Word clouds generated from the CoT outputs of LLaMA and Qwen on GSM8K.

| | Reason | Answer | Full |
|---|---|---|---|
| Original | 1070 | 287 | 1356 |
| Ours | 1495 | 310 | 1805 |

Table 12: Experimental results for ELI5 on Qwen under greedy decoding.

