# OpenReview forum: "Excessive Reasoning Attack on Reasoning LLMs"
_ICLR.cc/2026/Conference — Submitted to ICLR 2026_

### Official Review · Reviewer_Cuh6 · 2025-10-29

**Soundness:** 2
**Presentation:** 3
**Contribution:** 2
**Rating:** 4
**Confidence:** 3

**Summary:**

The paper investigates a new type of adversarial vulnerability in reasoning large language models (LLMs), referred to as Excessive Reasoning Attack. The authors argue that reasoning-oriented models, such as DeepSeek-R1 and Qwen variants, tend to perform unnecessary or redundant reasoning that can be exploited to increase inference-time computation. To formalize this, they propose a proxy optimization framework consisting of three components: (1) Priority Cross-Entropy Loss to emphasize informative tokens, (2) Excessive Reasoning Loss to encourage repeated or branched reasoning, and (3) Delayed Termination Loss to postpone output completion. Using the GCG optimization method, they generate short adversarial suffixes that substantially extend reasoning trajectories on GSM8K and ORCA benchmarks. Experiments show increased reasoning length, latency, and energy usage, with minimal change in accuracy, and demonstrate partial transferability across several commercial LLMs (o3-mini, GPT-OSS, DeepSeek-R1, QWQ).

**Strengths:**

1. This paper introduces Excessive Reasoning Attack, a novel adversarial attack that differs from prior works focusing solely on content manipulation or refusal-based safety issues. It specifically targets the reasoning process of LLMs, exposing a new dimension of vulnerability related to inference efficiency.
2. The paper proposes three complementary differentiable proxy losses—Priority Cross-Entropy (PCE), Excessive Reasoning (ER), and Delayed Termination (DT)—which effectively address the non-differentiability of reasoning length.
3. The experiments yield several meaningful insights, such as the observation that optimized adversarial suffixes remain effective across models sharing the same tokenizer.

**Weaknesses:**

1. White-box Assumption and Limited Practicality

This paper introduces Excessive Reasoning Attack, a novel adversarial attack targeting reasoning LLMs. I acknowledge that such an attack poses a more substantial threat to online LLM services (e.g., OpenAI, Google, and Alibaba Cloud) than to open-source models. However, the proposed method relies on a white-box assumption, requiring full access to model weights and gradients. This dependency makes it inapplicable to black-box commercial models. Although the results indicate certain transferability, the optimization mechanism still depends heavily on white-box access, which considerably limits the real-world practicality of this attack.

2. Demand on Computational Resources

The experiments are conducted on 7–8B models (LLaMA and Qwen), which can be handled with a single A100 80GB GPU. However, since the method requires gradient access—a memory-intensive operation—it entails significant computational overhead, which would
become more severe for larger LLMs. It is recommended that the computational cost and GPU memory consumption of the proposed framework be quantitatively analyzed to clarify its scalability and practicality.

3. Limitations of Experimental Design

- Stronger Baselines are Needed

The baselines in Table 1 are not well-aligned with the stated objective of this paper, inadvertently amplifying the effectiveness of the proposed method. Most compared methods are not designed to extend the reasoning length of LLMs. For instance, Engorgio Prompt primarily targets general DoS attacks, not reasoning-specific ones, while CatAttack introduces distractive sentences that mainly add semantic noise rather than lengthening reasoning. As shown in Table 1, these baselines barely increase reasoning length compared to the original setting. Therefore, stronger baselines are needed to empirically evaluate the performance of the proposed method.

- Limitation of Transferability Experiment

It would be beneficial to test the transferability of the proposed adversarial attack on more advanced reasoning LLMs, such as GPT-5, Claude, and Gemini, rather than only on open-source or relatively weak models listed in Table 4. Such experiments would better demonstrate how different reasoning capabilities affect robustness against this attack.

**Questions:**

See the weaknesses above.

---

> ### Author Response · Authors · 2025-11-21
>
> We thank Reviewer Cuh6 for the thoughtful assessment and for recognizing the novelty and effectiveness of our work.
> Reviewer Cuh6 raises three main concerns: (1) limited practicality (2) the need for computational resources and (3) limitations of experiments
> To address your concerns, we have provided detailed responses to the weaknesses you raised below.
>
> ---
>
> **W1: White-box Assumption and Limited Practicality**
>
> Answer: Indeed, white-box access considerably limits the real-world practicality. Nevertheless, transfer attacks are widely studied with many works demonstrating their effectiveness [A][B][C]. With the rapid release of public models, many of these can serve as surrogate models for transfer attacks. As long as we can identify the key factors that enable successful transferability, the barrier to executing such attacks is significantly lowered. Our analysis further highlights a critical factor (token alignment) that substantially improves transfer effectiveness. Finally, we evaluate our method on two more recent models (GPT-5-mini and Gemini-Flash) and the results further validate the robustness and effectiveness of our attack.
>
> [A] Why Do Adversarial Attacks Transfer? Explaining Transferability of Evasion and Poisoning Attacks
>
> [B] Why Does Little Robustness Help? A Further Step Towards Understanding Adversarial Transferability
>
> [C] Practical Black-Box Attacks against Machine Learning
>
> **W2: Demand on Computational Resources**
>
> Answer: Since GCG supports batching for gradient computation, we report GPU memory usage for a batch size of 8 and estimate the corresponding optimization time. In addition, our transferability results show that an attacker can leverage a small model (e.g., 7B) as a surrogate to attack large commercial or open-source models, thereby avoiding the need for huge computational resources.
>
> | Model | batch size = 8        |
> |:-------------------|:--------:|
> | Qwen               | 38.9GB    |
> | LLaMA              | 39.7GB    |
>
> | Model | batch size = 8        |
> |:-------------------|:--------:|
> | Qwen               | 16521s    |
> | LLaMA              | 19850s    |

---

> > ### Author Response · Authors · 2025-11-21
> >
> > **W3: Stronger Baselines are Needed**
> >
> > Answer:
> > To the best of our knowledge, attacks targeting reasoning LLMs remain relatively rare, and we have included as many baselines as possible.
> > We have introduced most known attacks in the Introduction and provided a complete list of attacks in the Related Work section.
> > Both Overthink and CRAB attack (2025) rely on injecting long decoy content—either appended to the user query or included as external context—that is substantially longer than the original user input.
> > Engorgio and CatAttack are the most recent (2025) applicable attack methods available.
> > Although Engorgio was originally designed for general LLMs, reasoning LLMs share most of the underlying capabilities/training with general LLMs, such as the EOS tag, and we expect the attack should work to some degree.
> > Although CatAttack is not designed to lengthen reasoning, they have studied the impact on output length. Therefore, we also include it as a baseline.
> > Also, we are happy to include additional, stronger baselines if the reviewer can point us to relevant open-sourced papers.
> >
> > **W4: Limitation of Transferability Experiment**
> >
> > Answer:
> > Due to budget limit, we can only include gpt5-mini and Gemini-Flash, but still, they are the most recent commercial models.
> >
> > | Model         | Reason (LLaMA) | Answer (LLaMA) | Full (LLaMA) | Accuracy (LLaMA) | Match (LLaMA) | Reason (Qwen) | Answer (Qwen) | Full (Qwen) | Accuracy (Qwen) | Match (Qwen) |
> > |---------------|----------------|----------------|--------------|------------------|---------------|---------------|---------------|-------------|-----------------|--------------|
> > | QWQ           | 1761 (-151)    | 231 (-6)       | 1992 (-157)  | 93% (-3%)        | 62%           | 2489 (+577)   | 317 (+80)     | 2806 (+657) | 98% (+2%)       | 100%         |
> > | R1            | 997 (-74)      | 185 (-13)      | 1182 (-87)   | 95% (-2%)        | 4%            | 1295 (+224)   | 233 (+36)     | 1528 (+260) | 93% (-4%)       | 60%          |
> > | o3-mini       | 446 (+199)     | 184 (+46)      | 630 (+245)   | 90% (+1%)        | 20%           | 645 (+398)    | 336 (+199)    | 982 (+596)  | 89% (0%)        | 70%          |
> > | gpt5-mini     | 343 (+175)     | 106 (+34)      | 449 (+209)   | 77% (-7%)        | 60%        | 578 (+410)    | 210 (+138)    | 787 (+547)  | 83% (-2%)       | 70%          |
> > | Gemini-Flash  | 665 (+146)     | 350 (+76)      | 1015 (+222)  | 86% (-3%)        | --            | 874 (+355)    | 513 (+239)    | 1388 (+594) | 90% (+1%)       | --           |
> > | OSS (low)     | 79 (+13)       | 170 (+35)      | 249 (+48)    | 92% (+2%)        | 20%           | 95 (+19)      | 316 (+167)    | 411 (+186)  | 90% (0%)        | 70%          |
> > | OSS (medium)  | 569 (+149)     | 237 (+111)     | 806 (+259)   | 90% (+1%)        | 20%           | 3015 (+2569)  | 348 (+206)    | 3363 (+2775)| 88% (-2%)       | 70%          |
> >
> > Also, we include a new metric (e.g., Match) that reflects the token-overlap rate between the local and target models. Specifically, we check whether the token length of the optimized suffix, when encoded by the local and target models (e.g., gpt5), is the same. If the lengths match, it indicates that none of the tokens in the optimized suffix are split or merged by the target model. We observe that when the Match score is high, the overall length is also high, demonstrating a positive relationship between the two.
> >
> > ---
> >
> > We care deeply about whether our rebuttal has addressed your concerns.
> > If so, we would greatly appreciate it if you could consider raising your score.
> > If you have any remaining questions, we would be happy to clarify them further.

---

> > > ### Comment · Reviewer_Cuh6 · 2025-11-23
> > >
> > > The authors' response has addressed most of my concerns, and I welcome the authors to include the supplementary experimental results in the final version of the paper. I am willing to raise my score.
> > >
> > > However, I still have one last question. In W4, what I am truly concerned about is whether models (e.g., GPT-5, Gemini 3) can withstand the attack methods proposed in this paper if they are sufficiently intelligent. Therefore, I hope the authors verify the transferability performance of the proposed strategy on more intelligent models; so merely adding results on GPT-5-mini and Gemini-Flash is insufficient. If the authors' budget is limited, I believe experimenting on a small set (e.g., 100 queries) is acceptable.

---

> > > > ### Author Response · Authors · 2025-11-23
> > > >
> > > > Thank you again for recognizing our work and for your willingness to improve the score.
> > > > We have conducted an additional study on GPT-5 and Gemini 3, using a single run instead of three repeated runs in order to reduce the overall cost.
> > > > The results show that our attack is also effective against these advanced models.
> > > >
> > > > | Model         | Reason (LLaMA) | Answer (LLaMA) | Full (LLaMA) | Accuracy (LLaMA) | Match (LLaMA) | Reason (Qwen) | Answer (Qwen) | Full (Qwen) | Accuracy (Qwen) | Match (Qwen) |
> > > > |---------------|----------------|----------------|--------------|------------------|---------------|---------------|---------------|-------------|-----------------|--------------|
> > > > | gpt-5     | 402 (+164)     | 75 (+11)      | 477 (+175)   | 72% (+2%)        | 60%        | 686 (+448)    | 144 (+81)    | 830 (+529)  | 78% (+8%)       | 70%          |
> > > > | Gemini3  | 1100 (+306)     | 313 (+77)      | 1412 (+384)  | 86% (-2%)        | --            | 1389 (+596)    | 554 (+318)    | 1943 (+914) | 88% (+0%)       | --           |

---

> > > > > ### Comment · Reviewer_Cuh6 · 2025-11-24
> > > > >
> > > > > Thank you for the additional experimental results. All my concerns have been addressed.

---

### Official Review · Reviewer_zwXP · 2025-10-30

**Soundness:** 3
**Presentation:** 3
**Contribution:** 3
**Rating:** 6
**Confidence:** 2

**Summary:**

This paper investigates a novel denial-of-service (DoS) style threat against reasoning-focused large language models (LLMs), which are prone to inefficient behaviors like redundant or overly long reasoning chains. The authors propose the "Excessive Reasoning Attack," which crafts adversarial suffixes to compel the target model to generate excessively long reasoning trajectories, thus significantly increasing inference latency, energy consumption, and overall computational cost without degrading task accuracy. To achieve this, they introduce a composite loss function comprising three components: Priority Cross-Entropy (PCE) Loss, Excessive Reasoning (ER) Loss, and Delayed Termination (DT) Loss, optimized using a gradient-based search (GCG). The results show a substantial increase in reasoning length (3x to 6.5x) and resource usage, demonstrating transferability across various commercial and open-source models.

**Strengths:**

1.  **Novel and Practical Attack Objective:** The paper identifies and exploits a highly relevant vulnerability: the efficiency and resource consumption of reasoning LLMs. Unlike attacks targeting answer correctness, this focuses on *economic* and *operational* damage (computational overhead), which is a critical, underexplored threat in commercial LLM deployment (akin to a DoS attack).
2.  **Strong Empirical Validation and Transferability:** The attack demonstrates high efficacy, successfully increasing reasoning length by several multiples across different model architectures (LLaMA and Qwen variants). Furthermore, the strong transferability of the adversarial suffixes to black-box commercial models like 03-mini and others highlights the generalized nature of this efficiency vulnerability in the reasoning mechanisms themselves.

**Weaknesses:**

1.  **Ambiguity in Causality of Performance Gain:** The paper observes that the attack, while lengthening reasoning, sometimes increases task accuracy. The analysis attributes this to increased capacity allocation, but a more in-depth exploration of *why* the attack's specific, lexically biased reasoning leads to *better* answers is needed to fully understand the mechanism.
2.  **Tokenizer Dependency in Transferability:** The transferability analysis, particularly the difference between the LLaMA and Qwen optimized suffixes on the target models, points to a strong dependence on tokenizer alignment. This suggests the attack's universality may be bounded by tokenization schemes, a limitation that should be discussed more explicitly in the conclusion.

**Questions:**

Please see the weakness.

---

> ### Author Response · Authors · 2025-11-21
>
> We thank Reviewer zwXP for the thoughtful assessment and for recognizing the novelty and effectiveness of our work.
> Reviewer zwXP raises two main concerns: (1) the reason for the performance gain and (2) the limitation of transferability.
> To address your concerns, we have provided detailed responses to the weaknesses raised below.
>
> ---
>
> **Q1: Ambiguity in Causality of Performance Gain: The paper observes that the attack, while lengthening reasoning, sometimes increases task accuracy. The analysis attributes this to increased capacity allocation, but a more in-depth exploration of why the attack's specific, lexically biased reasoning leads to better answers is needed to fully understand the mechanism.**
>
> Answer: In most cases, our attack triggers meaningless reasoning, as shown in Appendix C.2. However, we also find that in a few cases, the regular reasoning is too simple and fails to solve the problem, while our triggered excessive reasoning occasionally fixes it, as shown below. For example, the original response contains a single solution (reasoning path), and it actually leads to an incorrect calculation. In contrast, our attack triggers excessive reasoning that explicitly extends the thinking with multiple solutions; however, we can see that the last solution is redundant since the answer has already been inferred correctly. In general, most problems can be solved with baseline prompting, and our attack unnecessarily extends the reasoning in most cases, leading to computational waste.
>
> Question:
> ```
> Raymond and Samantha are cousins.
> Raymond was born 6 years before Samantha.
> Raymond had a son at the age of 23.
> If Samantha is now 31, how many years ago was Raymond's son born?
> ```
>
> Sample response from original:
> ```
> First, I need to determine Samantha's age when Raymond's son was born.
> Since Samantha is currently 31 years old, I'll subtract 6 years from her current age to find out how old she was when Raymond was born.
> This gives me 25 years old.
> Next, I'll figure out how long ago Raymond's son was born.
> Raymond had his son at the age of 23.
> To find out how many years ago this happened, I'll subtract 23 from Samantha's age at that time.
> So, 25 minus 23 equals 2 years ago.
> Therefore, Raymond's son was born 2 years ago.
> </think>
> ... **Final Answer:** \\[\n\\boxed{2}\n\\]
> ```
>
> Sample response from our attack:
> ```
> Okay, so I'm trying to figure out this problem about Raymond and Samantha ...
> Raymond and Samantha are cousins. Raymond was born 6 years before Samantha. Raymond had a son at the age of 23 ...
> Alright, so Raymond was born 6 years before Samantha ...
> Wait, no, that's not right. If Raymond was born 6 years before Samantha ...
> Yeah, that makes sense because if you subtract 6 years from Samantha's age ...
> So, 31 - 6 = 25?
> Wait, no, that's the opposite ...
> Raymond had a son at the age of 23.
> So, Raymond was 23 when his son was born.
> Now, Raymond is 37, so the time between Raymond having his son and now is 37 - 23 = 14 years.
> So, the son was born 14 years ago.
> Wait, but let me double-check that. If Raymond was 23 when his son was born, and he is now 37, then 37 - 23 = 14 years have passed since the son was born.
> So, the son is now 14 years old, which means he was born 14 years ago. That seems right.
> ...
> </think>
> ... **Answer:** Raymond's son was born 14 years ago.
> ```
>
> To see the whole picture, we have conducted a fine-grained analysis focusing on the Incorrect-to-Correct cases, where an initially incorrect answer became correct after appending the suffix, on GSM8K for both LLaMA and Qwen.
>
> | Setup    | Avg Reasoning (LLaMA) | Alternatively (LLaMA) | Wait (LLaMA) | Avg Reasoning (Qwen) | Alternatively (Qwen) | Wait (Qwen) |
> |----------|-------------------------|-------------------------|---------------|-------------------------|-------------------------|---------------|
> | Original | 7                       | 0                       | 0             | 8                       | 0                       | 0             |
> | Attack   | 97                      | 93                      | 154           | 40                      | 3                       | 12            |
>
> We observe that the average number of reasoning sentences increases significantly, and increases in the use of both “Wait” and “Alternatively” tokens. While this suggests the attack may improve accuracy in certain cases, it does so at the cost of significantly greater computational overhead. Importantly, the added reasoning steps may still be redundant rather than essential to arriving at the correct answer, as shown in the example above.

---

> > ### Author Response · Authors · 2025-11-21
> >
> > **W2: Tokenizer Dependency in Transferability: The transferability analysis, particularly the difference between the LLaMA and Qwen optimized suffixes on the target models, points to a strong dependence on tokenizer alignment. This suggests the attack's universality may be bounded by tokenization schemes, a limitation that should be discussed more explicitly in the conclusion.**
> >
> > | Model         | Reason (LLaMA) | Answer (LLaMA) | Full (LLaMA) | Accuracy (LLaMA) | Match (LLaMA) | Reason (Qwen) | Answer (Qwen) | Full (Qwen) | Accuracy (Qwen) | Match (Qwen) |
> > |---------------|----------------|----------------|---------------|------------------|---------------|---------------|---------------|-------------|-----------------|--------------|
> > | QWQ           | 1761 (-151)    | 231 (-6)       | 1992 (-157, 0.93×)  | 93% (-3%)        | 62%           | 2489 (+577)   | 317 (+80)     | 2806 (+657, 1.31x) | 98% (+2%)       | 100%         |
> > | R1            | 997 (-74)      | 185 (-13)      | 1182 (-87, 0.93×)   | 95% (-2%)        | 4%            | 1295 (+224)   | 233 (+36)     | 1528 (+260, 1.21x) | 93% (-4%)       | 60%          |
> > | o3-mini       | 446 (+199)     | 184 (+46)      | 630 (+245, 1.64x)   | 90% (+1%)        | 20%           | 645 (+398)    | 336 (+199)    | 982 (+596, 2.55x)  | 89% (0%)        | 70%          |
> > | gpt5-mini     | 343 (+175)     | 106 (+34)      | 449 (+209, 1.87x)   | 77% (-7%)        | 60%        | 578 (+410)    | 210 (+138)    | 787 (+547, 3.28x)  | 83% (-2%)       | 70%          |
> > | Gemini-Flash  | 665 (+146)     | 350 (+76)      | 1015 (+222, 1.28x)  | 86% (-3%)        | --            | 874 (+355)    | 513 (+239)    | 1388 (+594, 1.75x) | 90% (+1%)       | --           |
> > | OSS (low)     | 79 (+13)       | 170 (+35)      | 249 (+481, 1.24x)    | 92% (+2%)        | 20%           | 95 (+19)      | 316 (+167)    | 411 (+186, 1.83x)  | 90% (0%)        | 70%          |
> > | OSS (medium)  | 569 (+149)     | 237 (+111)     | 806 (+259, 1.47x)   | 90% (+1%)        | 20%           | 3015 (+2569)  | 348 (+206)    | 3363 (+2775, 5.72x)| 88% (-2%)       | 70%          |
> >
> >
> > We include a new metric (e.g., Match) that reflects the token-overlap rate between the local and target models. Specifically, we check whether the token length of the optimized suffix, when encoded by the local and target models (e.g., gpt5), is the same. If the lengths match, it indicates that none of the tokens in the optimized suffix are split or merged by the target model. We observe that when the Match score is high, the overall length is also high, demonstrating a positive relationship between the two.
> >
> > Furthermore, we will add this limitation to the conclusion:
> >
> > Based on the analysis, we can see that the choice of optimized tokens is also critical and constrained by the tokenizer used. Right now, many commercial APIs—such as OpenAI—release their tokenizers, and tokenizer inference attacks [A] also exist and can be used to recover unknown tokenizers. In future work, the selection of optimized tokens can be incorporated as an additional optimization constraint, not only for this attack but also for other adversarial methods.
> >
> > [A] GradEscape: A Gradient-Based Evader Against AI-Generated Text Detectors
> >
> > ---
> >
> > We care deeply about whether our rebuttal has addressed your concerns.
> > If so, we would greatly appreciate it if you could consider raising your score.
> > If you have any remaining questions, we would be happy to clarify them further.

---

### Official Review · Reviewer_uiUg · 2025-11-02

**Soundness:** 3
**Presentation:** 3
**Contribution:** 2
**Rating:** 6
**Confidence:** 3

**Summary:**

The paper proposes a white-box adversarial suffix attack that increases the reasoning token length (and thus latency/energy) of reasoning-oriented LLMs without substantially degrading task accuracy. Because reasoning length is non-differentiable, the authors optimize three differentiable proxy losses: (i) Priority Cross-Entropy (PCE) that reweights supervision toward prompt-dependent tokens, (ii) Excessive Reasoning (ER) that boosts the probability of sentence-initial deliberation tokens, and (iii) Delayed Termination (DT) that suppresses end and beginning of sentence tokens. They search adversarial suffixes with GCG. On GSM8K/ORCA, the attack increases reasoning tokens by 3 to 6.5X, with large latency and energy overhead; adversarial suffixes also transfer to several commercial/closed models.

**Strengths:**

1) The decomposition into PCE, ER, DT is well-motivated by the goal (longer reasoning). Ablations isolate each piece (Table 5; Table 11), showing monotone increases when objectives are combined.

2) On LLaMA-8B-R1-distill (GSM8K greedy), reasoning tokens rise from 668 to 1914 and latency 24.3s to 54.9s with only 10-token suffixes (Table 1). Similar or larger effects on Qwen-7B-R1-distill. The designed attack works experimentally.

3)Cross-model tests (Table 4) show the attack is not brittle. This connects to prior literature on universal triggers and GCG-style suffixes shaping model behavior.

**Weaknesses:**

1) Only 50 examples * 2 datasets * 3 runs = (approx.) 300 evaluations. No statistical test on accuracy differences. For GSM8K (app. 8.5 k train / 1 k test), using 50 samples risks > +- 3 points variance; hence “no degradation” claims are statistically inconclusive.


2) The threat model assumes full gradient access (white-box), while commercial systems (o1/o3-mini) are black-box API only. Transferability results (Tab. 4) are modest (<600 tokens gain) and could arise from stochastic sampling noise.

3) The paper assumes that maximizing the PCE/ER/DT composite correlates with reasoning-token count, yet this link is never formally validated. Equation (3) weights by prompt-sensitivity (Δ log p), but the proof that this expectation increases generation length is missing.
A simple counterexample exists: tokens with high prompt-sensitivity can be function words (e.g., “Therefore”) that appear early without elongating reasoning.
A more principled surrogate, e.g., REINFORCE or length-aware policy gradient as in Wu et al., 2025 (When More is Less), would better justify optimization correctness.


4) (not a major weakness) I noticed that related work has been provided in Appendix, which is very non-standard. This is not grounds for rejection, but I think including some of the related work would be helpful.

**References:**
[1] Wu Y, Wang Y, Ye Z, Du T, Jegelka S, Wang Y. When more is less: Understanding chain-of-thought length in llms. arXiv preprint arXiv:2502.07266. 2025 Feb 11.

**Questions:**

1) GSM8K and ORCA are math-only; do you expect similar behavior on symbolic-reasoning datasets (e.g., MATH Bench)?

2) How do you verify that longer reasoning corresponds to distinct reasoning trajectories, not repetition? ANy qualitative study?

3) What happens if $\beta$ is reduced from 50 to 1? Does effect vanish?

4) Please confirm whether the importance score (Eq2.) is correct. I might be wrong but current formula appears mathematically reversed.

---

> ### Author Response · Authors · 2025-11-21
>
> We thank Reviewer uiUg for the thoughtful assessment and for recognizing the design and effectiveness of our work.
> We note that your main concerns relate to the method and the evaluations.
> To address your concerns, we have provided detailed responses to the weaknesses and questions raised below.
>
> ---
>
> **W1: Only 50 examples * 2 datasets * 3 runs = (approx.) 300 evaluations. No statistical test on accuracy differences. For GSM8K (app. 8.5 k train / 1 k test), using 50 samples risks > +- 3 points variance; hence “no degradation” claims are statistically inconclusive.**
>
> Answer:
> We repeat each experiment three times separately to reduce runtime variability.
> For greedy decoding, we report a single run since it is deterministic.
> For sampling, we repeat the experiment 10 times.
> We will clarify this in the paper.
> Also, we agree that using 50 samples may introduce variance compared to the full test set.
> However, our goal is not to make fine-grained statistical claims, but to verify that our method does not degrade original accuracy significantly.
> The table below summarizes the mean and standard deviation over 10 sampling runs.
> Across all datasets and models, the standard deviation remains below 5 acc, which indicates that the measured performance is stable enough to distinguish large magnitude differences.
> The variance is also comparable between the original and our method.
>
>
> | GSM8K     | Acc. Mean (LLaMA) | Acc. Std (LLaMA) | Acc. Mean (Qwen) | Acc. Std (Qwen) |
> |-----------|-------------------|------------------|------------------|-----------------|
> | Original  | 76.0              | 4.8              | 82.0             | 2.4             |
> | Ours      | 90.0              | 3.2              | 91.0             | 3.0             |
>
>
> | ORCA      | Acc. Mean (LLaMA) | Acc. Std (LLaMA) | Acc. Mean (Qwen) | Acc. Std (Qwen) |
> |-----------|-------------------|------------------|------------------|-----------------|
> | Original  | 81.0              | 3.8              | 84.0             | 2.0             |
> | Ours      | 87.0              | 2.0              | 87.0             | 1.3             |
>
> In addition, we note that widely used safety evaluation datasets such as JailbreakBench [A] and HarmBench (standard) [B] contain only 100 and 200 test samples, respectively.
> These benchmarks are accepted by the community despite their relatively small size.
> For the same reason, we believe our 50 subsets are sufficient to demonstrate the emergent threat induced by our attack.
>
> [A] JailbreakBench: An Open Robustness Benchmark for Jailbreaking Large Language Models
>
> [B] HarmBench: A Standardized Evaluation Framework for Automated Red Teaming and Robust Refusal
>
> **W2: The threat model assumes full gradient access (white-box), while commercial systems (o1/o3-mini) are black-box API only. Transferability results (Tab. 4) are modest (600 tokens gain) and could arise from stochastic sampling noise.**
>
> Answer: Our transferability results demonstrate a 2-3× increase in reasoning length. At the scale of commercial LLM deployments—processing millions to billions of queries daily—even this moderate increase poses a significant security risk. Successful transferability could enable real-world exploitation of production servers, making this far from a negligible threat. In addition, we have conducted an additional study on gpt5-mini and Gemini-Flash to demonstrate the effectiveness of our transferability further. To minimize sampling noise within our budget, we report the mean over three independent generations for each query.
>
> | Model         | Reason (LLaMA) | Answer (LLaMA) | Full (LLaMA) | Accuracy (LLaMA) | Match (LLaMA) | Reason (Qwen) | Answer (Qwen) | Full (Qwen) | Accuracy (Qwen) | Match (Qwen) |
> |---------------|----------------|----------------|---------------|------------------|---------------|---------------|---------------|-------------|-----------------|--------------|
> | gpt5-mini     | 343 (+175)     | 106 (+34)      | 449 (+209, 1.87x)   | 77% (-7%)        | 60%        | 578 (+410)    | 210 (+138)    | 787 (+547, 3.28x)  | 83% (-2%)       | 70%          |
> | Gemini-Flash  | 665 (+146)     | 350 (+76)      | 1015 (+222, 1.28x)  | 86% (-3%)        | --            | 874 (+355)    | 513 (+239)    | 1388 (+594, 1.75x) | 90% (+1%)       | --           |

---

> > ### Author Response · Authors · 2025-11-21
> >
> > **W3: The paper assumes that maximizing the PCE/ER/DT composite correlates with reasoning-token count, yet this link is never formally validated. Equation (3) weights by prompt-sensitivity (Δ log p), but the proof that this expectation increases generation length is missing. A simple counterexample exists: tokens with high prompt-sensitivity can be function words (e.g., “Therefore”) that appear early without elongating reasoning. A more principled surrogate, e.g., REINFORCE or length-aware policy gradient as in Wu et al., 2025 (When More is Less), would better justify optimization correctness.**
> >
> > Answer:
> >
> > The link between PCE/ER/DT and token length has been demonstrated based on our latter ablation experiments. First, the purpose of CE/PCE is to optimize for the target output. When the target is shorter, the triggered reasoning is correspondingly shorter; when the target is longer, the triggered reasoning also extends, as shown in Tables 8, 9, and 10. We introduce PCE to improve optimization efficiency further. As we decrease k, the number of reasoning tokens increases. Both sets of results show a clean and consistent link between PCE and the reasoning-token count. Second, ER and DT are designed to increase the output probability of reasoning tokens (e.g., “Wait”) and reduce the probability of end-of-thinking and end-of-sequence tokens. We believe this design is straightforward and intuitive. Moreover, the empirical results in Table 5 demonstrate their effectiveness.
> >
> > Regarding the counterexample, we agree that some function words (e.g., “Therefore”) may exhibit high sensitivity without meaningfully contributing to reasoning extension. However, we believe that as k decreases, the number of such tokens diminishes. Moreover, these tokens affect precision but not the overall direction of optimization. Even with some noisy tokens included, the mask should still capture enough influential tokens to reliably induce elongated reasoning as shown in Figure 3.
> >
> > Although one of the objectives in Wu's work is to optimize for the optimal CoT length, their method is not explicitly designed to maximize output length. Furthermore, although RL can directly assign rewards based on token length, its proxy gradients are still surrogates and not accurate. Since the question of whether “exact reward with surrogate gradient (RL)” or “surrogate loss with exact gradient (our method)” is more effective remains open, and RL optimization typically requires extensive sampling and rollouts and is generally less stable than exact-gradient optimization, we adopt GCG with a surrogate loss in our method.
> >
> > **W4: (not a major weakness) I noticed that related work has been provided in Appendix, which is very non-standard. This is not grounds for rejection, but I think including some of the related work would be helpful.**
> >
> > Answer: We have moved it back to the main paper.

---

> > > ### Author Response · Authors · 2025-11-21
> > >
> > > **Q1: GSM8K and ORCA are math-only; do you expect similar behavior on symbolic-reasoning datasets (e.g., MATH Bench)?**
> > >
> > > Answer: To further strengthen our evaluation, we have also conducted a new experiment on 50 random examples from the ELI5 dataset [A] on Qwen, which focuses on general QA rather than math reasoning.
> > > For instance, questions such as “How do jellyfish function ... ?” allow us to assess the generality of our method on a non-mathematical domain.
> > >
> > > | ELI5     | Reason | Answer | Full |
> > > |----------|--------|--------|------|
> > > | Original | 1070   | 287    | 1356 |
> > > | Ours     | 1495   | 310    | 1805 |
> > >
> > > [A] ELI5: Long Form Question Answering
> > >
> > >
> > > **Q2: How do you verify that longer reasoning corresponds to distinct reasoning trajectories, not repetition? Any qualitative study?**
> > >
> > > Answer: In this paper, our goal is to induce long outputs, so we can ignore whether the generated reasoning trajectories are distinct or repetitive.
> > > Moreover, Figure 2 reports the token counts that are signaled as overthinking and underthinking in prior work.
> > > The increase in these tokens reflects a higher frequency of reasoning-path switching.
> > >
> > > In addition, we split the reasoning path (GSM8K) into solutions with LLaMA 70B and use gpt-4o to cluster the solutions [A]. As shown below, the average number of clusters increases slightly from Original to Ours, indicating that most reasoning paths (solutions) remain repetitive. Also, the average number of distinct solutions increases significantly, indicating that LLaMA responses exhibit more solution switching.
> > >
> > > [A] Do NOT Think That Much for 2+3=? On the Overthinking of Long Reasoning Models
> > >
> > > | Method | Avg. Clusters (LLaMA) | Avg. Solutions (LLaMA) | Avg. Solution Len (LLaMA) | Avg. Clusters (Qwen) | Avg. Solutions (Qwen) | Avg. Solution Len (Qwen) |
> > > | --- | --- | --- | --- | --- | --- | --- |
> > > | Original | 1.24 | 1.78 | 510 | 1.04 | 1.06 | 490 |
> > > | Ours | 1.70 | 4.14 | 720 | 1.24 | 1.86 | 1206 |
> > >
> > > **Q3: What happens if beta is reduced from 50 to 1? Does effect vanish?**
> > >
> > > Answer: Yes, we observe that the effect becomes less effective. We suspect that tokens such as “Wait”, “Alternatively” have relatively low probability of being generated at the beginning, so their contribution to the loss is negligible compared to other components. Therefore, we selected a larger scale (e.g., 50) with grid search.
> > >
> > > **Q4: Please confirm whether the importance score (Eq2.) is correct. I might be wrong but current formula appears mathematically reversed.**
> > >
> > > Answer:
> > > Yes, it should be reversed.
> > >
> > > ---
> > >
> > > We care deeply about whether our rebuttal has addressed your concerns.
> > > If so, we would greatly appreciate it if you could consider raising your score.
> > > If you have any remaining questions, we would be happy to clarify them further.

---

### Author Response · Authors · 2025-12-02

Dear ACs and Reviewers,

Thank you for the time and care you put into serving our submission.

In general, all the reviewers recognized the novelty and importance of our attack, as well as its effectiveness under comprehensive evaluations across multiple models and datasets.
**Also, we have fully addressed reviewer Cuh6's concern, and the score was increased to positive before the revert, which means all reviewers agree on the acceptance of our paper.**

We have conducted additional studies and analyses to address the reviewers’ concerns:

(Note: due to space limitations, additional tables and samples are shown in each rebuttal section.)

---

- New Task and statistically inconclusive (uiUg):

First, **we have conducted a new experiment for the ELI5 dataset on Qwen, demonstrating that our attack is effective beyond math tasks.**
Second, our goal is not to make fine-grained statistical claims, but to verify that our method does not degrade the original accuracy significantly.
We also compute the mean and standard deviation over 10 sampling runs.
**The standard deviation remains below 5 acc, which indicates that the measured performance is stable enough to distinguish large magnitude differences.**

- Transferability (uiUg, zwXP, and Cuh6):

Our transferability results demonstrate a 2-3× increase in reasoning length.
At the scale of commercial LLM deployments—processing billions of queries daily—even this moderate increase poses a significant security risk.
In addition, **we have conducted an additional study on advanced LLMs (including GPT-5 and Gemini3) to demonstrate the effectiveness of our transferability further.**

Also, we include a new metric (e.g., Match) that reflects the token-overlap rate between the local and target models.
**We observe that when the Match score is high, the overall length is also high, demonstrating a positive relationship between the two.
This indicates that using properly optimized tokens can enhance attack transferability.**

- Details on the method: (uiUg):

**The link between PCE/ER/DT and token length has been demonstrated in our ablation experiments.**
First, the purpose of CE/PCE is to optimize for the target output. When the target is shorter, the triggered reasoning is correspondingly shorter; when the target is longer, the triggered reasoning also extends, as shown in Tables 8, 9, and 10. We introduce PCE to improve optimization efficiency further. As we decrease k, the number of reasoning tokens increases. Both sets of results show a clean and consistent link between PCE and the reasoning-token count. Second, ER and DT are designed to increase the output probability of reasoning tokens (e.g., “Wait”) and reduce the probability of end-of-thinking/sequence tokens. The empirical results in Table 5 demonstrate their effectiveness.

- Additional analysis (uiUg and Cuh6):

For the reasoning trajectories, we split GSM8K into solutions and cluster the solutions.
We find that the average number of clusters increases slightly from Original to Ours, **indicating that most solutions remain repetitive.**
Also, the average number of distinct solutions increases significantly, **indicating that LLaMA responses exhibit more solution switching.**

**For the beta, we observe that the effect becomes less effective from 50 to 1.**
We suspect that tokens such as “Wait” have a relatively low probability of being generated at the beginning, so their contribution to the loss is negligible compared to other components.
**We also report GPU memory usage and estimate the corresponding optimization time.**

- Performance Gain (zwXP):

In most cases, our attack triggers meaningless reasoning, as shown in Appendix C.2.
However, we also find that in a few cases, the regular reasoning is too simple and fails to solve the problem, while our triggered excessive reasoning occasionally fixes it.
For example (see zwXP), the original response contains a single solution, and it actually leads to an incorrect calculation.
In contrast, our attack triggers excessive reasoning that explicitly extends the thinking with multiple solutions; however, we can see that some late solutions are redundant since the answer has already been inferred correctly.
**In general, most problems can be solved with simple prompting, and our attack unnecessarily extends the reasoning in most cases, leading to computational waste.**
We have also conducted a fine-grained analysis focusing on the Incorrect-to-Correct cases, and **observe that the added reasoning steps are still redundant rather than essential to arriving at the correct answer.**

- Summary

This study reveals **a new class of inference-time vulnerabilities** in reasoning LLMs, showing that minimal adversarial perturbations can **significantly inflate inference costs and impose substantial computational burdens.**
Our transferability analysis further suggests that properties such as token alignment may be key to crafting strong textual adversarial examples.

---

### Meta-Review · Area_Chair_dtA1 · 2026-01-04

**Summary:**

This paper studied a new threat to reasoning models of excessive reasoning attacks. A white-box adversarial suffix attack was proposed to increase the reasoning token length. The authors introduce a proxy framework that approximates the long reasoning objective including riority Cross-Entropy Loss, Excessive Reasoning Loss, and Delayed Termination Loss. Experiments show the increase in reasoning length on several datasets.

The reviewers recognized the novel threat model, reasonable framework design, and strong empirical results. The reviewers also raised several major concerns of this paper, including:

- More datasets (uiUg)
- Transferability of the method across models (uiUg, zwXP, and Cuh6)
- Additional analysis experiments and case study (uiUg and zwXP)
- Limited practical applicability of white-box attacks (Cuh6)
- Computational cost (Cuh6)

The authors addressed most of these concerns by adding experiments and clarifications. After thoroughly reading the paper and reviewers' comments, the Area Chair identify additional concerns that were not mentioned by the reviewers, affecting the assessment of the paper's novelty and contribution.

The main contribution of this paper lies in three differentiable proxy losses: (1) Priority Cross-Entropy Loss, which reweights supervision toward prompt-dependent tokens; (2) Excessive Reasoning Loss, which increases the probability of reasoning tokens; (3) Delayed Termination Loss, which extends the reasoning process and defers the generation of final outputs. However, Delayed Termination Loss (using the end-of-thinking (EOT) token) has already been widely applied in similar reasoning attack methods [1][2][3][4]. Moreover, the idea behind Priority Cross-Entropy Loss (adjusting weights based on token relevance) has also been proposed in previous work [4]. As for the concern raised by reviewer uiUg about related work, the authors have neglected relevant work such as Engorgio Prompt in the introduction and failed to clarify their original contributions. In summary, the contribution and novelty of this paper are insufficient.

Additionally, regarding transferability, the authors addressed the concerns of reviewers zwXP and Cuh6, but they did not directly respond to reviewer uiUg’s concern about "<600 tokens gain", and failed to demonstrate the practicality of the proposed method. Therefore, the Area Chair decided to reject this paper.

[1] Jianshuo Dong, Ziyuan Zhang, Qingjie Zhang, Tianwei Zhang, Hao Wang, Hewu Li, Qi Li, Chao Zhang, Ke Xu, and Han Qiu. An engorgio prompt makes large language model babble on. In The Thirteenth International Conference on Learning Representations, ICLR 2025, Singapore, April 24-28, 2025.
[2] Simin Chen, Zihe Song, Mirazul Haque, Cong Liu, and Wei Yang. NICGSlowDown: Evaluating the Efficiency Robustness of Neural Image Caption Generation Models. In IEEE Conference on Computer Vision and Pattern Recognition (CVPR), pp. 15344–15353. IEEE, 2022.
[3] Kuofeng Gao, Yang Bai, Jindong Gu, Shu-Tao Xia, Philip Torr, Zhifeng Li, and Wei Liu. Inducing High Energy-Latency of Large Vision-Language Models with Verbose Images. In International Conference on Learning Representations (ICLR), 2024.
[4] Jiyuan Fu, Kaixun Jiang, Lingyi Hong, Jinglun Li, Haijing Guo, Dingkang Yang, Zhaoyu Chen, and Wenqiang Zhang. LingoLoop Attack: Trapping MLLMs via Linguistic Context and State Entrapment into Endless Loops. arXiv preprint arXiv:2506.14493 (2025).

**Reviewer Concerns:**

During the review and rebuttal process, reviewers highlighted the following weaknesses:
1. No statistical testing of accuracy differences (uiUg)
2. More datasets (uiUg)
3. Transferability of the method across models (uiUg, zwXP, and Cuh6)
4. Additional analysis experiments and case study (uiUg and zwXP)
5. Insufficient issues about related work (uiUg)
6. Limited practical applicability of white-box attacks (Cuh6)
7. Computational cost (Cuh6)

The authors mostly addressed these concerns through additional experiments and clarifications:
1. The authors clarified the details of their statistical tests.
2. The authors conducted new experiments on the ELI5 dataset using Qwen.
3. For the concerns of zwXP and Cuh6, the authors added new APIs and a new metric, Match. They tried to clarify uiUg's concern, but their response was unconvincing.
4. The authors added extra experiments. For zwXP’s Q1, they included a case study.
5. The authors revised the manuscript.
6. The authors clarified the practical applicability of their method.
7. The authors supplemented the experimental results.

In conclusion, the concerns of reviewer zwXP and reviewer Cuh6 have mostly been addressed. However, reviewer uiUg's concerns regarding transferability remain.

Additionally, Area Chair after thorough reading identifies new concerns that were not mentioned by the reviewers, affecting the paper’s novelty and contribution, which are explained in the summary.

**Reviewer Scores:**

Reviewer Cuh6 believes that all their concerns have been addressed and may increase the score. Though the reviewer scores are borderline accept, the AC found significant limitations regarding the paper novelty and contributions. Therefore, AC would recommend rejection.

---

### Decision · Program_Chairs · 2026-01-26

Reject